https://doi.org/10.1038/s41467-022-29267-8　**OPEN**

# Congruent evolutionary responses of European steppe biota to late Quaternary climate change

Philipp Kirschner [1,2,12 ✉], Manolo F. Perez [3,4,12], Eliška Záveská[1,5], Isabel Sanmartín [3], Laurent Marquer [1], Birgit C. Schlick-Steiner[2], Nadir Alvarez[6,7], the STEPPE Consortium*, Florian M. Steiner[2,13] & Peter Schönswetter[1,13 ✉]

Quaternary climatic oscillations had a large impact on European biogeography. Alternation of cold and warm stages caused recurrent glaciations, massive vegetation shifts, and large-scale range alterations in many species. The Eurasian steppe biome and its grasslands are a noteworthy example; they underwent climate-driven, large-scale contractions during warm stages and expansions during cold stages. Here, we evaluate the impact of these range alterations on the late Quaternary demography of several phylogenetically distant plant and insect species, typical of the Eurasian steppes. We compare three explicit demographic hypotheses by applying an approach combining convolutional neural networks with approximate Bayesian computation. We identified congruent demographic responses of cold stage expansion and warm stage contraction across all species, but also species-specific effects. The demographic history of the Eurasian steppe biota reflects major paleoecological turning points in the late Quaternary and emphasizes the role of climate as a driving force underlying patterns of genetic variance on the biome level.

[1] Department of Botany, University of Innsbruck, Sternwartestraße 15, 6020 Innsbruck, Austria. [2] Department of Ecology, University of Innsbruck, Technikerstraße 25, 6020 Innsbruck, Austria. [3] Real Jardín Botánico, CSIC, Plaza de Murillo 2, 28014 Madrid, Spain. [4] Departamento de Genetica e Evolucao, Universidade Federal de Sao Carlos, Rodovia Washington Luis, km 235, 13565905 Sao Carlos, Brazil. [5] Institute of Botany of the Czech Academy of Sciences, Zámek 1, 25243 Průhonice, Czech Republic. [6] Geneva Natural History Museum of Geneva, Route de Malagnou 1, 1208 Genève, Switzerland. [7] Department of Genetics and Evolution, University of Geneva, Boulevard D'Yvoy 4, 1205 Genève, Switzerland. [12] These authors contributed equally: Philipp Kirschner, Manolo F. Perez. [13] These authors jointly supervised this work: Florian M. Steiner, Peter Schönswetter. *A list of authors and their affiliations appears at the end of the paper. ✉email: philipp.kirschner@gmail.com; peter.schoenswetter@uibk.ac.at

The recurrent alternation of cold (glacial periods) and warm stages (interglacial periods) during the Quaternary (the last 2.6 million years, myr) was paramount in shaping present-day species distribution patterns in Europe. Transitions were marked by large fluctuations of temperature and precipitation occurring within millennia[1,2] and fueled extensive range expansions and contractions in many biota[3]. Phylogeography has contributed significantly to our knowledge about the impact of these climatic fluctuations on the European flora and fauna[4,5]. Pleistocene sea-level changes, glacier advances, and retreats, as well as the complex topography of Europe—with high mountain chains such as the Alps and the Pyrenees acting as major dispersal barriers—seemingly led to large-scale extinction within some groups (e.g., the Tertiary tree flora[6]), while promoting the formation of novel evolutionary entities in other groups via recurrent isolation[7–9].

The Eurasian steppe is a biome that has undergone massive climate-driven contractions and expansions in the Quaternary. Today, it extends over several thousand kilometers, from the northern coast of the Black Sea in Ukraine in the west throughout Central Asia to northwestern China in the east[10]. Low annual precipitation is the decisive factor preventing the formation of closed forests, which renders steppe grasslands the zonal (i.e., macroclimatically induced) vegetation in these areas[11]. During the cold stages of the Pleistocene, such as the last glacial period (LGP), 115 to 12 kya, the Eurasian steppe had a much larger extent, compared with interglacial periods[12–14], and repeatedly expanded into large parts of present-day, forest-covered temperate Europe[15–17] (Fig. 1). In present-day temperate Europe, isolated patches of steppe vegetation, the so-called extrazonal steppes, occur apart from the zonal steppes, resembling steppe islands in a sea of (potential) forest[18]. These extrazonal steppes occur wherever local factors, such as southern exposition and shallow soil cover, act in concert with a continental climate to prevent forest growth[12]; Fig. 1).

Extrazonal steppes have traditionally been considered remnants of an extensive, continuous cold stage steppe belt[14] (Fig. 1B), which became isolated from each other and from the zonal steppe due to postglacial forest expansion at the start of the Holocene, 11,700 years ago[12]. However, Kirschner et al.[19] recently reported that the isolation of the extrazonal steppe biota is in fact much older; vicariant separation occurred as early as the mid-Pleistocene, c. 1.4 mya. Range contractions triggered by forest expansion recurrently forced Eurasian steppe biota into disjunct and—compared with their extent during glacial stages—small-sized interglacial refugia (i.e., the present-day extrazonal steppes), since the very onset of the Pleistocene.

Climate fluctuations likely drove large demographic changes over time in both extrazonal and zonal steppes. Yet, the genetic signatures resulting from these processes are not fully understood.

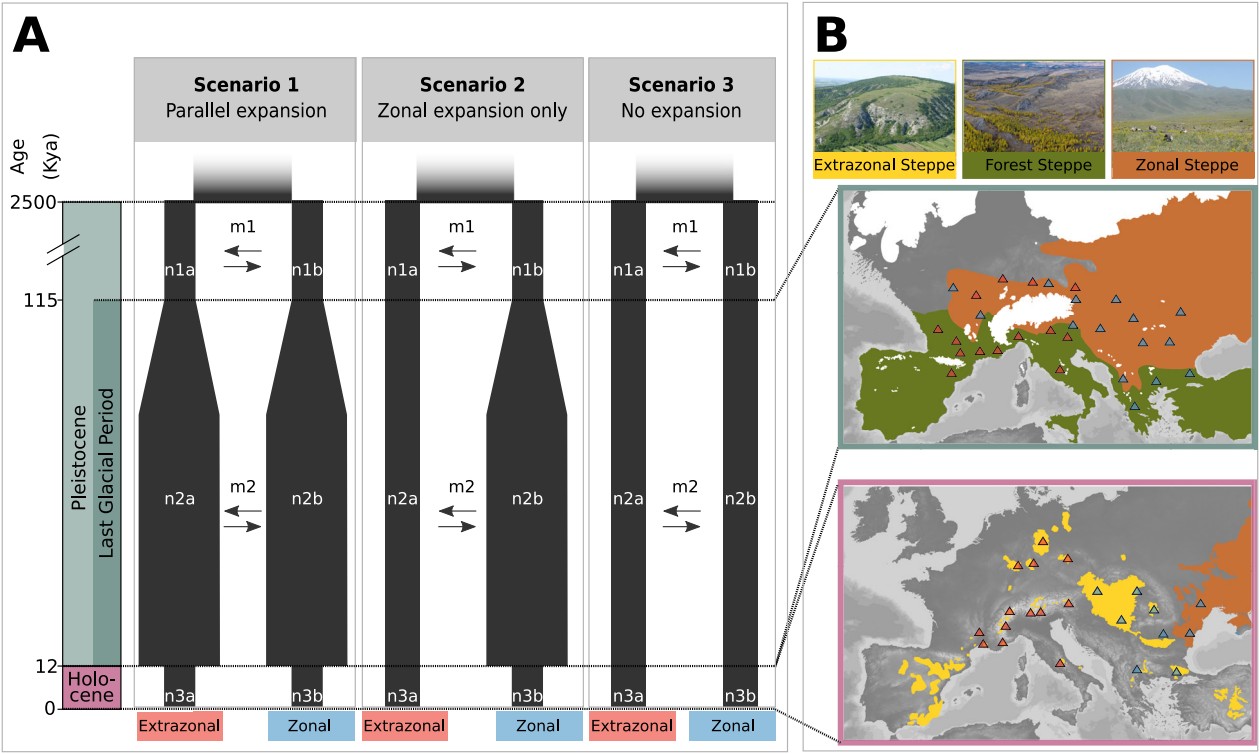

**Fig. 1 Hypothetical scenarios for the demography of steppe biota in the light of late Quaternary steppe range dynamics. A** Overview of the three demographic scenarios evaluated for the European steppe biota in relation to age and geological epoch (scaling not continuous). In Scenario 1, Parallel expansion, an increase in effective population size (*n*) in extrazonal and zonal steppe lineages during the last glacial period (LGP), followed by a contraction of *n* in the Holocene, is assumed. Scenario 2, Zonal expansion only, implies expansion and contraction for zonal steppe lineages, but a constant population size for extrazonal steppe lineages. Scenario 3, No expansion, involves no change in effective population size in neither extrazonal nor zonal lineages. All scenarios include bidirectional migration (*m*) between the extrazonal and zonal steppe lineages in the pre-LGP and LGP periods. **B** Vegetation in Europe under climatic conditions of the Last Glacial Maximum ~21,000 kya (upper map, modified from Anhuf et al.[77]), and under present-day (that is, warm stage) conditions (modified from Kirschner et al.[19]; Wesche et al.[10]) brown areas represent zonal steppe (i.e., macroclimatically induced continuous steppe), yellow areas represent extrazonal steppes (patchy steppe occurrences embedded in a matrix of the forest), green areas represent forest steppe (open forest with an understory of steppe biota), and white areas show glaciated areas[78]. Blue and red triangles show hypothetical occurrences of populations of steppe species pertaining to extrazonal and zonal lineages, respectively.

An obvious demographic scenario is that climate-driven range expansions led to demographic expansion in both extrazonal and zonal steppe populations during the LGP, followed by a demographic contraction in the Holocene (Scenario 1, Parallel expansion in Fig. 1A). This first scenario captures classical hypotheses of steppe expansion in Europe during the LGP[12,14]. Under an alternative scenario, demographic expansion took place only in zonal steppe populations but not in the extrazonal ones (Scenario 2, Zonal expansion only in Fig. 1A). In this case, the mountain barriers surrounding many extrazonal steppes, as well as their isolation due to forest spread during warm stages (Fig. 1B), would have prevented demographic expansion in extrazonal steppe lineages. A third scenario implies an absence of demographic expansion in both zonal and extrazonal steppe lineages, as a result of slow range shifts in the zonal steppe during the LGP (Scenario 3, No expansion in Fig. 1A).

Model-based statistical approaches allow a comparison of alternative demographic scenarios in terms of their fit to the data, and the inference of relevant parameters to explain patterns of genetic variation across geographic space while incorporating the uncertainty in parameter estimation[20]. One of the most popular approaches is approximate Bayesian computation (ABC), a flexible likelihood-free statistical framework based on simulations[21]. The ABC framework allows researchers to incorporate a-priori information about relevant parameters that are used to simulate genetic datasets under alternative demographic scenarios. The simulated data are then compared with the empirically observed data, using genetic summary statistics to discriminate among scenarios[22]. Recently, machine-learning approaches such as convolutional neural networks (CNN) have emerged as an alternative to ABC methods[23]. CNN can recover information directly from raw genetic datasets by converting them into images, thus overcoming the necessity to select a particular set of statistics to reduce the high dimensionality in the genetic data that affects traditional ABC methods[24]. Some recent studies have suggested that improved accuracy can be achieved by combining these two methods, that is, by using machine-learning CNN predictions as an input to perform ABC parameter estimation[25].

In this study, we applied a statistical modeling approach based on a coupled CNN and ABC framework to five Eurasian steppe species, three insects and two angiosperms. Inferences were based on genomic sequence data obtained via restriction-site associated DNA sequencing (RADseq) by Kirschner et al.[19]. Within each of these species, geographic isolation of two genetic groups, an extrazonal lineage and a zonal lineage, reflecting their main distribution in either zonal or extrazonal steppes, was demonstrated[19]. Using pairwise comparisons of zonal and extrazonal lineages within this phylogenetically diverse array of species, we aimed to test which of the three scenarios outlined above (Parallel expansion, Zonal expansion only, No expansion) shows a better fit to the late Quaternary population-size dynamics of European steppe biota. We then estimated relevant demographic parameters for the selected scenarios, such as effective population sizes, divergence times, migration rates, and timing of expansion/contraction events. Finally, we evaluated congruence in demographic responses across species, using independent palynological and paleoclimatic evidence, as well as hindcasted distribution models reflecting the climatic niche of each species' extrazonal and zonal lineages during the Last Glacial Maximum (LGM; c. 21,000 y ago). Here, we show that our CNN approach clearly identifies models capturing demographic expansions (Parallel expansion & Zonal expansion only) during the LGP as the best fitting evolutionary scenarios in all five phylogenetically distant study species. The initial splits between zonal and extrazonal lineages as well as the onset of demographic expansions in the LGP correspond to significant turning points in the

Quaternary period and are reflected in climatic and palynological data. Consequently, we argue that the revealed climate-driven dynamics reflect a general pattern that applies to many European steppe biota.

## Results

**Clustering analyses**. Population grouping into two clusters, corresponding to extrazonal and zonal lineages, was the optimal solution for all investigated species based on Bayesian population assignment (Fig. 2A). Though Kirschner et al.[19] found further subgrouping within these clusters, the focus of our study is the contrasting demographic dynamics between extrazonal and zonal steppe lineages, so we constrained all analyses to the clusters at the highest hierarchical level, that is, $K = 2$ for all species. The number of individuals and the number of single nucleotide polymorphisms (SNPs) analyzed are given in Table 1.

Geographic distribution of, and degree of admixture between, the extrazonal and zonal lineages were found to be species-specific (Fig. 2A). The highest level of admixture was found in populations north of the Alps (*Euphorbia seguieriana*, *Stipa capillata*) and in the Pannonian basin (*E. seguieriana*, *Plagiolepis taurica*). In addition, a few admixed populations were found in the Western Alps (*E. seguieriana*, *P. taurica*, *Stenobothrus nigromaculatus*, *S. capillata*). Populations from the Apennines show evidence of admixture in two species, *P. taurica* and *S. capillata*; for the latter, an amphi-Adriatic disjunction was found within the zonal lineage, which was not observed in any other species.

**Divergence time estimation**. Estimates of $\tau$ ($\tau = 2\mu t$; $\mu$ mutation rate per site per generation, $t$ divergence time) were generally consistent among the differently sized alignments (Supplementary Fig. 1). The initial divergence between extrazonal and zonal lineages was estimated to have occurred in or after the mid-Pleistocene across all species analyzed. These estimations of divergence times were based on $\tau$ inferred from the largest subset (500 RADseq loci). Estimates of absolute divergence times and highest posterior density credibility intervals (HPD) were 0.59 mya (95% HPD: 0.34–0.86 mya) for *E. seguieriana*, 1.39 mya (95% HPD: 1.11–1.7 mya) for *O. petraeus*, 1.07 mya (95% HPD: 0.60–1.60 mya) for *P. taurica*, 0.38 mya (95% HPD: 0.29–0.46 mya) for *S. nigromaculatus*, and 0.8 mya (95% HPD: 0.46–1.12 mya) for *S. capillata*.

**Exploratory demographic analyses**. For most species, stairway plots suggested stable effective population sizes in both extrazonal and zonal lineages during the last 100 ky, followed by a decline of population size between 10 and 20 kya, which marks the end of the LGP (Supplementary Fig. 2). Deviations from this pattern are observed in *E. seguieriana*, for which a stable population size through time was inferred. Population size increases at around 100 kya were found in the zonal and extrazonal lineages of *E. seguieriana*, and in the extrazonal lineage of *P. taurica* and *O. petraeus*; a similar pattern, but somewhat earlier, was observed in *S. nigromaculatus* (Supplementary Fig. 2). This result is biologically plausible and concurs with the onset of the LGP. We refrained from interpreting more ancient population size changes, as artificial signals may occur near the method's lower inference limit[26,27].

**CNN based demographic modeling**. Our combined CNN–ABC approach (Fig. 3) for selecting the best-fit demographic scenario resulted in high overall model accuracy for all study species. The cross-validation procedure, using a test set of simulations not evaluated during the training step, gave a percentage of correct

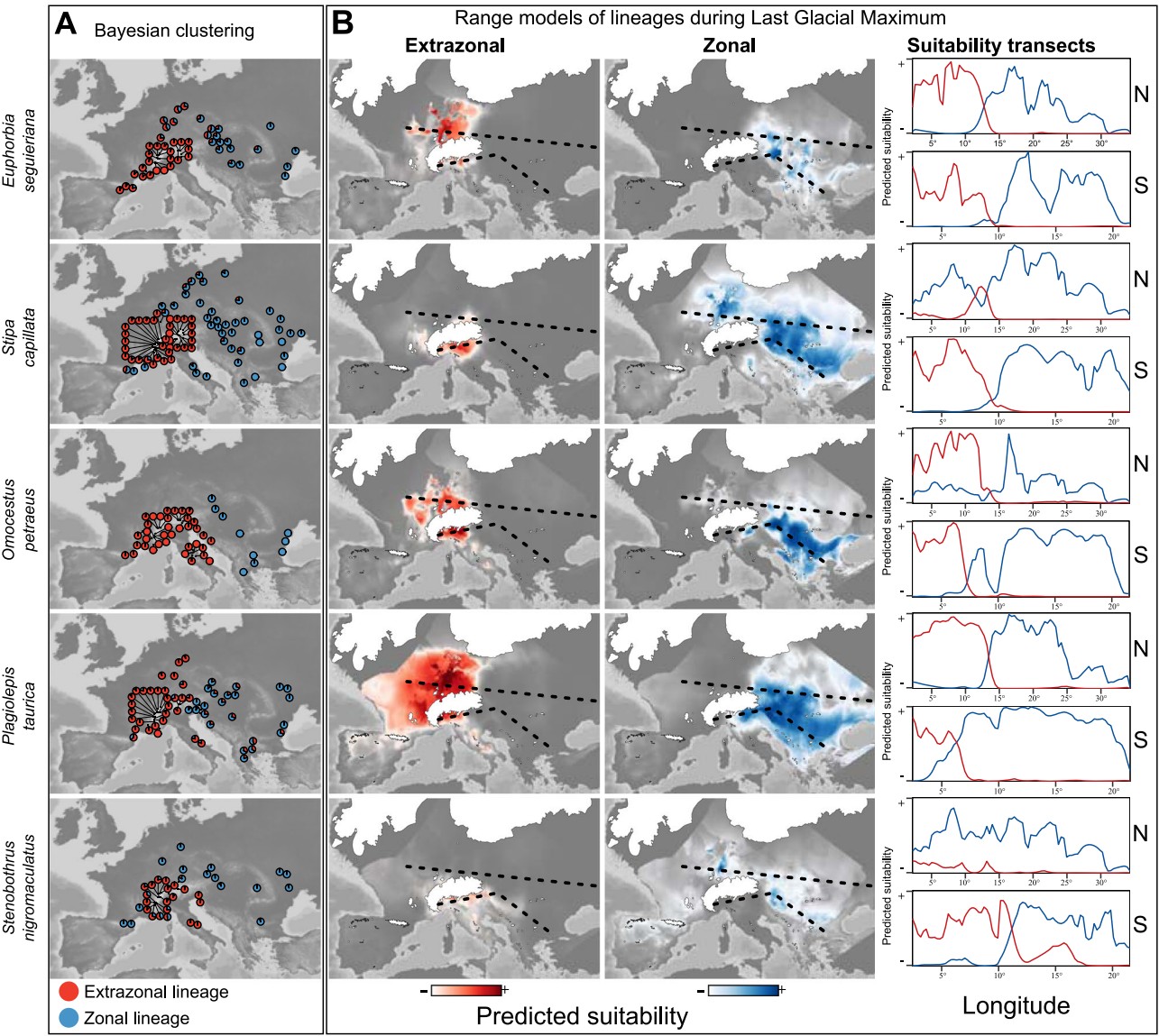

**Fig. 2 Extant distribution of genetic lineages and modeled past ranges. A** Distribution of extrazonal (red) and zonal (blue) lineages of steppe species based on genetic clusters inferred via Bayesian clustering. Each pie chart represents a population. **B** Distribution of predicted species-specific habitat suitability (non-standardized) during the Last Glacial Maximum (21 kya) based on niche models using climatic variables. Suitability transects show predicted habitat suitability for the extrazonal lineage (red line) and the zonal lineage (blue line) along the dashed lines depicted in the maps; x-axes represent longitude; N, northern transect; S, southern transect. Source data are provided as a Source Data file.

assignment to the simulated scenario higher than 65% in *P. taurica*, 75% in *O. petraeus* and 80% in all other species. Further, the calibration procedure improved the trained models, resulting in lower loss values (Supplementary Table 1). Our training strategy proved also to be effective for avoiding a loss of accuracy in SNP datasets that contain large amounts of missing data (e.g., in *S. nigromaculatus* and *O. petraeus*). The scenario depicting Parallel expansion in zonal and extrazonal lineages was selected as the most explanatory demographic model with a posterior probability (PP) higher than 0.99 in the angiosperm species *S. capillata* and the two grasshopper species *S. nigromaculatus* and *O. petraeus*. For *P. taurica* and *E. seguieriana*, the Zonal expansion only scenario was selected as the best model, with a PP value higher than 0.96. In *P. taurica*, model selection was not affected by using different generation times. The No expansion scenario showed a very low PP value across all species analyzed (Supplementary Table 2). Parameter estimation with CNN-ABC estimated large population sizes for *E. seguieriana*, *O. petraeus*, and

*P. taurica*, while *S. capillata* had the smallest values. We also inferred lower contemporary population sizes for the extrazonal compared with the zonal lineages across all analyzed species. Migration rates between zonal and extrazonal lineages within each time period were species-specific, with higher values during the LGP observed in *E. seguieriana* and *O. petraeus*, higher pre-LGP values estimated in *S. capillata*, and similar values for the two periods in the remaining species, *P. taurica* and *S. nigromaculatus* (Fig. 4). We did not infer common patterns of migration asymmetry between extrazonal and zonal lineages across the analyzed species.

**Distribution models for extrazonal and zonal lineages.** For the zonal lineages of each analyzed species, lineage distribution models (we use this term instead of lineage range model for readability purposes) suggested the existence of continuous distribution ranges, extending from the Pontic plains north of the

**Table 1 Overview of the number of individuals (CNN, Stairway Plot, BPP, and STRUCTURE) and the number of single nucleotide polymorphisms (SNPs; CNN, Stairway Plot, STRUCTURE) used for the respective analysis and for each species (details in "Methods").**

| Species | CNN | | | Stairway plot | | | | BPP | | STRUCTURE | |
|---|---|---|---|---|---|---|---|---|---|---|---|
| | Lineage | Individuals | SNPs (% of missingness across individuals) | Lineage | Individuals | SNPs | L | Lineage | Individuals | Individuals | SNPs |
| *Euphorbia seguieriana* | ExZon | 80 | 12,125 (15%) | ExZon | 120 | 5623 | $5 \times 10^5$ | ExZon | 15 | 138 | 30,804 |
| | Zon | 135 | | Zon | 84 | 9122 | $8.119 \times 10^5$ | Zon | 15 | | |
| *Omocestus petraeus* | ExZon | 10 | 1763 (42%) | ExZon | 24 | 1964 | $1.748 \times 10^5$ | ExZon | 23 | 158 | 7016 |
| | Zon | 10 | | Zon | 12 | 1213 | $1.08 \times 10^5$ | 1Zon | 10 | | |
| *Plagiolepis taurica* | ExZon | 23 | 12,542 (17%) | ExZon | 64 | 4825 | $4.294 \times 10^5$ | ExZon | 18 | 142 | 23,825 |
| | Zon | 22 | | Zon | 29 | 7016 | $6.244 \times 10^5$ | Zon | 15 | | |
| *Stenobothrus nigromaculatus* | ExZon | 12 | 2922 (41%) | ExZon | 16 | 1068 | $0.95 \times 10^5$ | ExZon | 15 | 97 | 3088 |
| | Zon | 6 | | Zon | 9 | 1513 | $1.347 \times 10^5$ | Zon | 8 | | |
| *Stipa capillata* | ExZon | 102 | 3813 (27%) | ExZon | 30 | 4943 | $4.399 \times 10^5$ | ExZon | 15 | 262 | 9073 |
| | Zon | 98 | | Zon | 56 | 1828 | $1.627 \times 10^5$ | Zon | 15 | | |

L (Stairway Plot) refers to the total number of nucleic sites (both polymorphic and monomorphic sites) from which the SNPs were inferred. BPP analyses were based on full sequences of random subsets of RADseq fragments.

Black Sea to the Pannonian Basin east of the Alps during the late Quaternary cold stages (Fig. 2B). In *S. capillata* and *S. nigromaculatus*, geographic ranges reached further west along the northern margin of the Alps, into Germany and France. Extensive suitable areas south of the Alps were found only for *O. petraeus* and *P. taurica*. Towards the west, these ranges did not reach further than northeasternmost Italy.

Lineage distribution models for the extrazonal lineages of each species supported continuous ranges south of the Alps for all studied species. Large continuous ranges north of the Alps were modeled for *P. taurica*, and to a lesser extent for *E. seguieriana* and *O. petraeus*. Small potential ranges along the northern margin of the Alps were also inferred for *S. capillata*. Gaps in habitat suitability were found mainly south of the Alps (Fig. 2B). In this area, range overlap of extrazonal and zonal lineages was observed only for *P. taurica* and *S. nigromaculatus*. In contrast, north of the Alps, range overlap of extrazonal and zonal lineages occurred within each species, except in *O. petraeus*.

### Discussion

A classic hypothesis about the Quaternary range dynamics of European steppe species is that they responded in exactly the opposite way to the well-investigated European temperate forest biota[28]; that is, a climate-driven interplay of warm-stage (including the Holocene) range contractions and cold-stage range expansions[13]. Here, we demonstrated that the demographic responses of five ecologically similar, but distantly related, steppe species are in line with this hypothesis and seem to have been largely driven by climate fluctuations. However, demographic patterns were not strictly congruent across species, at least for the extrazonal lineages. Whereas large-scale expansions in both extrazonal and zonal lineages during the late Quaternary cold stages (Parallel expansion, Fig. 1) were supported in three species (*O. petraeus*, *S. nigromaculatus*, *S. capillata*; Fig. 4), a scenario without population expansion in the extrazonal lineage (Zonal expansion only) was inferred in the other two species (*E. seguieriana*, *P. taurica*; Fig. 4).

The congruent signal of demographic expansion observed in zonal steppe lineages across all study species (Fig. 4) agrees well with the hindcasted lineage distribution models during Quaternary cold stages, and with the pattern of climate-driven expansion of Eurasian steppes supported by palynological and paleoclimatic data (Fig. 2). Conversely, the pattern of no demographic expansion during cold stages exhibited by the extrazonal lineages of *E. seguieriana* and *P. taurica* (the Zonal expansion only scenario in Fig. 1) seems counterintuitive, given the large-scale availability of climatically suitable habitat during the LGP in the hindcasted models (Fig. 2B). Smaller population sizes (as predicted by the center-periphery hypothesis for peripheral populations[29]) and stronger substructuring of source populations previous to expansion, as well as the presence of mountain barriers preventing effective dispersal[30–33], were likely key factors that hindered range expansion and subsequent increases in effective population size in the extrazonal lineages, but less so in the zonal lineages. We emphasize that intrapopulation structure may also affect the ability of demographic methods to detect population expansion[34].

In addition, species-specific factors such as dispersal ability are known to affect the demographic response of a population to range expansion[31]. The two grasshopper species *O. petraeus* and *S. nigromaculatus* and the epizoochorous graminoid species *S. capillata*, all considered effective dispersers, seem to have followed the Parallel expansion scenario. In contrast, *E. seguieriana* and *P. taurica*, which supported a Zonal expansion only scenario, exhibit a more limited dispersal ability; this may be explained by a relatively large seed size and seed dispersal via myrmecochory in the plant species[35], and by small body size in the ant species[36]. Thus, a species' capacity for long-range dispersal probably played a role in the observed pattern of disconnected extrazonal steppes, but was less important in the more continuous zonal steppe ranges.

Our results support a timeline for the demographic history of genetic lineages in European steppe species during the late Quaternary that was roughly congruent across all study species (Fig. 4). For *O. petraeus* and *P. taurica*, we estimated divergence times for the initial split between the extrazonal and zonal lineages within each species that fall within the 95% HPD credibility intervals based on dated mitochondrial DNA (mtDNA) phylogenies[19]; the fact that our RADseq-based mean age estimates are on average younger may be explained by incomplete lineage sorting during initial divergence and/or inaccuracy in the implemented clock rate prior.

## A  Convert simulated SNPs to image

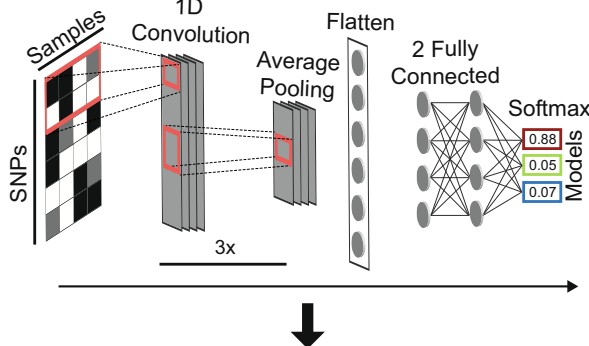

## B  Train CNN with images from all models

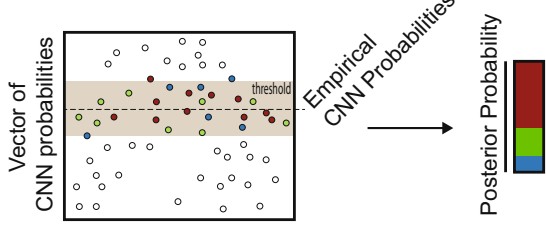

## C  Use CNN probabilities as ABC input

**Fig. 3 Graphic representation of the implemented convolutional neural network-approximate Bayesian computation (CNN-ABC) approach for model selection. A** We converted the simulated genetic data (10,000 simulations per scenario) to images, using the lowest value (−1; black pixels) for the reference state, the highest value (1; white pixels) for the alternative state, and an intermediate value (0; gray pixels) for the missing genotypes, which were randomly added according to the percentage of missing data observed in each species. **B** Each obtained image is then fed into a CNN that is trained to recognize images generated by simulations from each scenario (details on the CNN architecture in the Supplementary Information). **C** The vectors of CNN probabilities obtained for simulations from each scenario and for the empirical dataset are then used as summary statistics to perform an ABC step, by using a threshold that retains only the most similar simulations.

Calibration of genetic divergences to absolute time scales suggests that the timing of events in our demographic scenarios is related to periods that are considered climate turning points based on palynological evidence. We demonstrate this for three specific time horizons. Initial divergence between extrazonal and zonal lineages was estimated to have occurred between 0.37 and 1.39 mya by the Bayesian multispecies coalescent model implemented in BPP[37] and between 0.9 and 1.6 mya by CNN modeling (Supplementary Table 3, Fig. 4). These estimates roughly fall within a period known as the mid-Pleistocene

transition (1.25–0.7 mya), when the 41 ky glacial–interglacial cycles changed to 100 ky cycles[38]. In this period, an increase in the duration of glacial periods (c. 80–85 ky), compared with interglacial periods (c. 15–20 ky), likely favored the expansion of the steppe biota over a large part of Europe. We argue that the extended duration of warm stages during the mid-Pleistocene transition led to equally prolonged range contractions for the European steppe species, which likely facilitated initial allopatric divergence between today's extrazonal and zonal lineages. A similar pattern of intraspecific divergence during the mid-Pleistocene, referred to as the Pleistocene species pump has been found in European butterfly species, also on the basis of genome-wide data[8].

Our CNN models inferred a Late Pleistocene demographic expansion between 51 and 82 kya across all study species (Fig. 4, Supplementary Table 3). This period was characterized by a significant decrease in global mean temperatures[39] (Fig. 4) and corresponds to the marine isotopic stages 4 and 3, with a gradual cooling during the LGP, in particular during stage 4[40]. This colder climate likely triggered range expansions in European steppe species, which is also seen in the palynological record (Fig. 4, Supplementary Fig. 3). Pronounced demographic responses on a global scale have been inferred during this period in organisms with contrasting habitats[41,42], highlighting the severity and pervasiveness of Late Pleistocene climate change. Such congruence in demographic events across ecologically divergent species was interpreted as a direct effect of an abrupt global temperature drop induced by the eruption of the Toba supervolcano c. 74 kya[43]. Irrespective of the cause of this global event, we hypothesize that the rapidly cooling climate was key to massive demographic expansion in the European steppe biota.

Finally, a sharp decline in population size was inferred for all analyzed species around the mid to late Holocene (6.7–3.2 kya, Fig. 4). Interestingly, our data suggest that populations did not collapse immediately after the end of the LGP, ~12 kya, but during or after the Holocene climatic optimum (9–5 kya). The warm and humid climate during this period fostered the expansion of deciduous forests[28,44] and, at the same time, led to a decline of the remaining European steppes and forest steppes (Fig. 1). We conclude that the expansion of closed forest vegetation during the Holocene climate optimum was likely the final killing blow for many populations of the European steppe biota, triggering a rapid collapse in population size (Fig. 4, Supplementary Fig. 3).

Lineage distribution models for LGM conditions suggested large and continuous suitable habitats for both extrazonal and zonal lineages (Fig. 2B). Given that steppes were zonal—that is, microclimatically driven—vegetation under cold stage conditions, climate-based niche models likely well reflect the species' actual ranges at the LGM. This is less the case for niche models inferred for present-day, warm stage conditions. While present-day models are certainly restricted to areas with at least moderately continental climate[19], the actual occurrence of steppes within these modeled niches is largely determined by biotic interactions, specifically the lack of a dense forest cover. Modeling biotic interactions has proven problematic at the available spatial resolution[44]; we thus refrained from directly comparing the extent of warm stage and cold stage niches in the context of demography and lineage formation.

Lineage distribution models based on climatic variables indicate adjacency or even overlap of the LGM ranges of extrazonal and zonal lineages in all five study species to the north of the Alps (Fig. 2B). This is mirrored by the location of putative contact zones, where hybridization between extrazonal and zonal lineages has caused the frequent occurrence of admixed populations (*E. seguieriana*, *P. taurica* and *S. capillata*, Fig. 2). To the south of

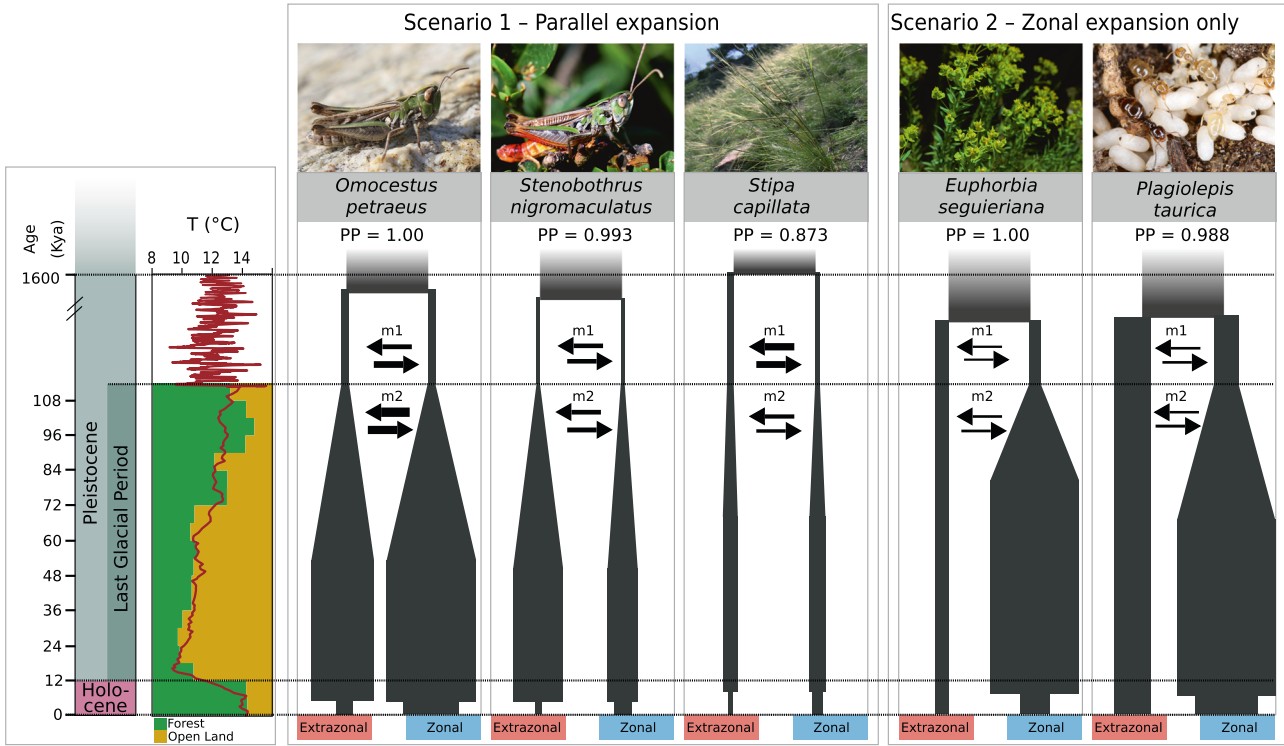

**Fig. 4 Best fitting demographic scenarios for the five study species.** Results showing selected demographic scenarios and associated parameters using convolutional neural network (CNN) modeling in five species endemic to the European steppes. Each species comprises an extrazonal (red) and a zonal (blue) lineage. The left inset figure shows the time scale with geological epochs and temperature fluctuations during the late Quaternary[39]: forest (green) and open land (yellow) pollen percentages throughout the last glacial period (LGP) are shown as bar plots; each bar represents 6000-year averages of the pollen percentages from five major pollen records covering the entire last glacial–interglacial cycle (data from European Pollen Database and PANGEA database, details in Supplementary Information)[15–17,79,80]. In each demographic scenario, the timing of the initial split between the zonal and extrazonal lineages, the LGP expansion, and the Holocene contraction are proportional to the time scale. The five species are assigned to their corresponding best-fitting model. Posterior probabilities (PP) values supporting the best-fitting model for each species are given above the demographic layout. In each layout, the diameters of branches and arrows are proportional to the inferred effective population sizes ($n$), and the inferred migration rates ($m$), respectively. For better visualization, branch diameters representing LGP population size were scaled down by 1/10 compared to pre- and post-LGP population sizes. Terminology for estimated parameters in Fig. 1; more detailed results in Supplementary Table 3.

the Alps, some range overlaps were also modeled, but large suitability gaps clearly prevail (Fig. 2B). Admixed populations are virtually absent (Fig. 2A); if they indeed existed, they were likely extirpated as the climate became unsuitable in the Holocene.

The dynamic oscillations of steppe vegetation in Europe during the late Quaternary are reflected in the modeled population size changes of the study species. Specifically, population expansions and contractions retrieved by our models were massive and in a similar scale across the investigated species (a 62- to 72-fold LGP increase in extrazonal lineages, 55- to 92-fold in zonal lineages; a 49- to 72-fold Holocene decrease in extrazonal lineages, 16- to 34-fold in zonal lineages; Fig. 4). CNN modeling also suggests that postglacial population contractions were more pronounced in the extrazonal than in the zonal steppe lineages, in agreement with the proportion of past and present availability of suitable habitat in Europe in these two groups (Figs. 1 and 2B). In other words, zonal lineages were able to maintain larger population sizes compared with extrazonal lineages because they had larger continuous ranges throughout the studied time periods (Fig. 1).

Simulation-based, likelihood-free modeling approaches, such as ABC or CNN, have become popular in phylogeography because of the ease to explore complex demographic scenarios with multiple interacting parameters without the need to derive the likelihood function of parameter dependencies[20,22]. However, these approaches may be less efficient for parameter estimation than full likelihood-based methods, such as Maximum Likelihood

or Bayesian inference, because they rely on simulations to explore a potentially broad range of parameter values[45,46]. Machine-learning CNN offers the advantage over ABC methods that information is extracted directly from the entire alignment of SNPs, better capturing patterns of genetic variation in genome-wide sequence data than the use of a single or multiple summary statistics[23,47]. In our study, we showed the power of a combined approach, in which CNN is used first to recover information directly from the SNP matrices and to reduce the initial parameter space, followed by an ABC rejection step (Fig. 3) based on CNN predictions[25]. The flexibility of this approach allowed us to similarly analyze datasets that were remarkably differing in size, dimension, and amount of missing data (Table 1). Our CNN-ABC approach allowed us to disentangle the demographic histories of a diverse array of distantly related European steppe species which differ in their ecology, dispersal mode, and reproductive strategy. In essence, we uncovered a congruent signal of climate-driven changes in geographic range and population sizes in zonal steppe lineages but idiosyncratic genetic histories for extrazonal lineages that might be linked to species-specific differences in effective dispersal distance.

## Methods

**Sampling.** Samples from 48 and 92 populations of two plant species belonging to different angiosperm families (the spurge *Euphorbia seguieriana* from Euphorbiaceae and the grass *Stipa capillata* from Poaceae, respectively) and samples from

56, 37, and 60 populations of three arthropod species from different insect orders (the grasshoppers *Omocestus petraeus* and *Stenobothrus nigromaculatus* and the ant *Plagiolepis taurica*, respectively) were included in this study. All samples were collected over the years 2013–2016 in mainly the western parts of the Eurasian steppes (Fig. 2A, Supplementary Data 1). The sampled species are all typical elements of the Eurasian steppe biome, and represent different reproductive, life-history, and dispersal strategies. Collecting permits are given in Kirschner et al.[19].

**Restriction-site associated DNA sequencing**. The RADseq data analyzed in this manuscript were generated by Kirschner et al.[19] using the original RADseq protocol[48] with minor modifications[49]. These data consist of 89 base pair single-end sequences that are available from the NCBI short read archive (Supplementary Data 1). From these data, genomic SNPs were called anew, using the version 2.3 of the Stacks package[50]. Several runs of denovo_map.pl were done on a subset of raw sequence data to optimize loci yield for each species, following Paris et al.[51]. The following species-specific parameters were used eventually: *E. seguieriana*, -n 3 -M 3 -m 5; *P. taurica* -n 3 -M 3 -m 5; *O. petraeus* -n 8 -M 8 -m 5; *S. nigromaculatus* -n 3 -M 3 -m 5; *S. capillata* -n 8 -M 8 -m5 (-n number of mismatches allowed between fragments between individuals), -M (number of mismatches allowed per fragment) and -m (minimum depth of coverage required to call a fragment)[50].

**Bayesian clustering analysis**. STRUCTURE v. 2.3.4[52] was used to explore patterns of genetic grouping within our datasets. Input files were exported from the Stacks catalog using the function populations.pl[50]. Only a single SNP per RADseq fragment (–write-single-snp flag in populations.pl) was selected to avoid linked SNPs, which violate the algorithm's assumption that SNPs are in linkage disequilibrium. In an additional filtering step, loci with excess heterozygosity (>65%) (–max-obs-het flag) were removed. This procedure has been suggested as a way to mitigate calling of paraloguous loci from RADseq data[53]. The final alignments contained only loci present in at least 40% (*E. seguieriana*), 15% (*O. petraeus*), 50% (*P. taurica*), 25% (*S. nigromaculatus*), or 33% (*S. capillata*) of all populations. STRUCTURE[52] was run assuming a grouping into $K = 1$ to 5 clusters for 1,000,000 generations, using a burnin of 100,000 generations and ten replicates per $K$. The optimal $K$ was assessed based on the rate change in likelihood among runs[54].

**Estimation of divergence times**. Relative divergence times between the extrazonal and zonal genetic lineages were inferred by applying a multispecies coalescent (MSC) model as implemented in the software BPP v. 4.2.9[37], and using the fixed topology approach (option A00). For each species, RADseq tags were exported from the Stacks catalog via populations.pl using the–fasta-samples flag and the –max-obs-het flag to remove loci with excess heterozygosity[50] (>65%, see also above), and were further converted from fasta files to phylip files using the python script fasta2genotype.py[55]. RADseq tags missing in more than 85% (*P. taurica*), 75% (*E. seguieriana*, *S. capillata*), or 50% (*O. petraeus*, *S. nigromaculatus*) were removed in each species. To reduce computational time, random subsets were generated containing 30 (*E. seguieriana*), 33 (*O. petraeus*), 33 (*P. taurica*), 30 (*S. capillata*), or 23 (*S. nigromaculatus*) individuals proportionally sampled from the extrazonal and zonal group in each instance (Table 1). Similarly, the full alignments of each species were randomly subsetted into smaller alignments containing 300, 400 and 500 RADseq loci for the final analysis. Analyzing SNP subsets of different sizes has been suggested as a way to evaluate the consistency of estimates within a given dataset[37,56,57]. The BPP analyses were run under default settings, assuming data to be diploid and unphased[37]. MCMC chain length was set to 1,000,000 generations, and 10% of the samples were discarded as burnin. All runs were checked in Tracer v. 1.6.0[58] to evaluate chain convergence to stationarity and adequate mixing, and to check if the effective sample size for estimated parameters reached at least 200.

Next, the function msc2time.r implemented in the R package bppr[59,60] was used to calibrate the relative branch lengths obtained in BPP to absolute divergence times. Specifically, this function calculates absolute divergence times based on MSC-derived estimates of $\tau$, by sampling mutation rate and generation time from a gamma distribution to obtain estimates of the mutation rate per absolute time. Mutation rates for each species were taken from literature-based estimates for genome-wide mutation rates (plants: 7e−9 substitutions per site per generation[61]; animals: 2.8e−9 substitutions per site per generation[62], and a deviation of 10% was allowed when calibrating $\tau$.

The generation times used to calibrate the time estimates were defined as the average time between two successive generations within a lineage or population[63]. In the case of the univoltine grasshopper species *O. petraeus* and *S. nigromaculatus*, this generation time is one year. For the ant species *P. taurica*, no species-specific data are available. The most thorough study on generation times in ants, targeting the red harvester ant *Pogonomyrmex barbatus*, found generation times (as defined above) in wild populations to be 7.8 years on average[64]. Here, two generation times were used to assess the robustness of CNN-informed model selection in *P. taurica*; specifically, a generation time of 10 ± 2 years as suggested by independent calibration from mtDNA-based phylogenetic inference and a shorter generation

time of 3 years that was perceived as biologically plausible considering the species small colony size, polygyny, and colony foundation via budding. Generation time estimates were not available for the two studied plant species, and the maximum lifespans of related and ecologically similar species were the only available references. Consequently, generation times of 10 ± 2 years and 25 ± 5 years were used for *E. seguieriana*[65] and *S. capillata*[66] (Podgaevskaya & Zolotareva pers. comm. 2020), respectively.

**Exploration of demographic history**. SNP data were exported to vcf files from the Stacks catalog using the–vcf flag in populations.pl, allowing for a single SNP per locus by using the –write-single-snp flag[50]; separate vcf files were generated for the extrazonal lineage and the zonal lineage (*n* of individuals and sites given in Table 1). From each of these variant files, individuals with an excessive amount of missing data were discarded, and the software vcftools[67] was subsequently used to remove SNPs that were missing in more than 85% (*E. seguieriana*), 75% (*P. taurica*, *S. capillata*), and 50% (*O. petraeus*, *S. nigromaculatus*) of the individuals. Calculation of the joint site frequency spectrum (SFS) was done using a custom Python script written by Isaac Overcast (available at GitHub https://github.com/isaacovercast/easySFS). This method is particularly suitable for RADseq data, as it handles missing data in the SNP matrix by down projecting to smaller sample size and averaging over all possible resamplings. Following the author's suggestions, down projection was chosen to retain the maximum number of individuals while avoiding the loss of too many SNPs.

The resulting SFS were used to explore population-size changes for each species and for each genetic lineage, using Stairway plots[26,27]. The blueprint files informing the algorithm were modified for each species, accordingly. Random breakpoints were defined as suggested[27]. Average generation times and mutation rates were the same as those used for divergence time estimation (see above). The remaining input parameters were not changed from the default settings.

**Demographic model testing using CNN**. To better understand and compare the demographic dynamics of each study species, we evaluated the three potential scenarios for the evolution of the European steppe biota during the Pleistocene climatic oscillations described in the Introduction (Parallel expansion, the Zonal expansion only, No expansion; Fig. 1). The number of individuals analyzed per species and lineage, and the number of SNPs are given in Table 1. We performed 10,000 coalescent simulations per scenario with the software ms[68], with species-specific priors for generation time and mutation rate as described above, and population sizes based on the Stairway plot results. Because our empirical SNP datasets included different levels of missing data (Table 1), we randomly inserted similar proportions of missing characters to the simulated SNP matrices for each species (Table 1, Fig. 3). This procedure allowed us to train the CNN to recover information from the genotype matrices, while also recognizing missing data. We used a network architecture (Fig. 3) based on Oliveira et al.[69], modified to include suggestions from Sanchez et al.[25], namely the use of different kernel sizes and intercalation of convolutional layers with batch normalization. The trained networks were then calibrated using temperature scaling[70] and used to predict the most likely model on the empirical SNPs and on a new set of 10,000 independent simulations per scenario. We also predicted parameter values for the empirical SNPs and 10,000 independent simulations for the preferred scenario. The obtained CNN predictions were then used to perform an ABC step with an optimized threshold level selected after trial runs (Fig. 3; with an approach similar to Mondal et al.[71]; and also recommended by Sanchez et al.[25]; for details, see the Supplementary Information).

**Distribution models for extrazonal and zonal lineages**. The potential range occupancy of extrazonal and zonal lineages under climatic conditions of the LGM was estimated using the lineage range estimation method[72]. Species distribution models under LGM climatic conditions based on two general circulation models (MIROC[73]; CCSM4[74]) were available for all study species[19]; lineage ranges of extrazonal and zonal lineages within each species were based on these models. Affiliation of each population to the extrazonal or the zonal lineage was derived from the STRUCTURE results; admixed populations were affiliated using a simple majority rule. Lineage range estimation followed the method by Rosauer et al.[72], using the R script provided by the authors (github.com/DanRosauer/phylospatial) with default parameters. A relaxed 10th percentile training presence (p10) threshold was applied. This approach was chosen because more stringent thresholds, such as the maximum training sensitivity plus specificity threshold, have been shown to severely under-represent species ranges if a niche is projected from a contracted present-day niche model, which is the case for the Eurasian steppe biota[75]. The suitability values of lineage distribution models were assessed along two transects north and south of the Alps, using the Temporal/Spectral Profile Tool v. 2.0.3 in QGIS v. 3.10. This gradient analysis was done to visualize continuities and gaps of habitat suitability within areas north and south of the Alps, which acted as a major distribution barrier for many species.

**Reporting summary**. Further information on research design is available in the Nature Research Reporting Summary linked to this article.

## Data availability

RADseq data are available from the NCBI GenBank Short Read Archive (accession numbers in Supplementary Data 1). Source data underlying Fig. 2 and Supplementary Figs. 1–3 are provided as Source Data files in a Figshare repository https://doi.org/10.6084/m9.figshare.19107944.v1 Source data are provided with this paper.

## Code availability

All scripts used to perform the presented CNN and ABC approaches are available at https://github.com/manolofperez/CNN_ABCsteppe[76].

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

## Acknowledgements

We thank all colleagues listed as collectors in Supplementary Data 1 that provided samples and also all colleagues that supported us by sharing locality data. We thank C. Lebas for providing his image of *P. taurica* shown in Fig. 4 and H. Wiesbauer for his aerial image of an extrazonal steppe shown in Fig. 1. The presented study was funded by the Austrian Science Fund (FWF, project P25955 Origin of steppe flora and fauna in inner-Alpine dry valleys to P.S.). M.F.P. was supported by the São Paulo Research Foundation (FAPESP) grant BEPE 2019/27089-8. We thank the center for Italian studies at the University of Innsbruck that supported our sampling campaign in Italy. We acknowledge the excellent HPC infrastructure LEO at the University of Innsbruck and also thank the Vienna Scientific Cluster (VSC)—both facilities were central for the success of this study.

## Author contributions

P.S., I.S., M.F.P., and P.K. designed the study. E.Z., M.F.P., and P.K. analyzed the data. M.F.P. conceived the CNN method. F.M.S., P.S., I.S., M.F.P., and P.K. co-wrote the paper. L.M. provided paleoecological expertize and data and wrote corresponding parts of the manuscript. B.C.S.-S. and N.A. contributed to the development of the manuscript and improved early drafts of the paper. Members of the Steppe Consortium contributed in manuscript writing and sample collection and provided lab expertise.

## Competing interests

The authors declare no competing interests.

## Additional information

**the STEPPE Consortium**

Wolfgang Arthofer[2], Božo Frajman[1], Alexander Gamisch[8], Andreas Hilpold[9], Philipp Kirschner [1,2,12✉], Ovidiu Paun[10], Isabel Sanmartín [3], Birgit C. Schlick-Steiner[2], Peter Schönswetter[1,13✉], Florian M. Steiner[2,13], Emiliano Trucchi[11] & Eliška Záveská[1,5]

[8]Department of Biosciences, University of Salzburg, Hellbrunnerstrasse 34, 5020 Salzburg, Austria. [9]Institute for Alpine Environment, Eurac Research, Drususallee 1, 39100 Bozen, Italy. [10]Department of Botany and Biodiversity Research, University of Vienna, Rennweg Vienna, Austria. [11]Department of Life and Environmental Science, Marche Polytechnic University, Ancona, Italy.

