## [Peer Review File · Nature Communications]

Reviewers' Comments:

Reviewer #1:

Remarks to the Author:

This is an excellent manuscript about the evolutionary responses of the European steppe biota to the Pleistocene climate changes which, as evidenced by the results, have produced deep effects on this biome. Given the relevance of the issue, and the conclusions presented, the ms. merits publication in Nature Communications. In addition, this ms. is complementing well a first paper also published in Nat. Comm. about its conservation value.

In my opinion, the ms. is novel and well-focused, with appropriate methods, and sound, well-discussed results. Albeit I am not familiar with the CNN methodology, the ABC is well implemented, with appropriate pre-defined scenarios. In addition to provide a few minor comments, my only major comment is concerning the distribution models (see below).

1. Lines 122-130. These lines are clearly Material & Methods, not Results.
2. Lines 227-231. I agree that small population sizes is a factor that can explain the lack of range expansion in the extrazonal lineages. The center-periphery hypothesis (CPH) predicts small population sizes for edge, peripheral populations.
3. Lines 351-353. Although the genetic structure will likely not change, I suggest to increase the number of replicates.
4. I am curious why you used only the LGM time slice to build the distribution models, as in recent times simulated paleoecological data for other Pleistocene time periods are available. These would be useful given the divergence time estimations between lineages. For example, data hosted on PaleoClim (<http://www.paleoclim.org/>) allow you to reconstruct the paleodistribution for the Last Interglacial and MIS19 (ca. 0,79 Mya), while with Oscillayers (<https://doi.org/10.5061/dryad.27f8s90>), time slices can be selected every 10,000 years.

Reviewer #2:

Remarks to the Author:

This paper provides an interesting further set of analyses, based on data generated by Kirschner et al, previously published in Nature Communications. The scope of the project is ambitious: different taxa from a biome, with the aim of understanding the demographic responses to past fluctuations in climate.

The analyses presented in this paper use fairly recently developed approaches in population genomic analysis. I have to say, from a genomic perspective, although the ambition is impressive, the actual data are relatively limited, and I have some doubts about aspects of these analyses.

I note that there is very little direct information in the main text, or the methods, about the extant of the data available. This is given in Supp Table 1. I suggest that it should be more upfront. To augment this, I would think some information on contig/scaffold lengths should be made available. For example BPP, as I understand it, is designed for short non-recombining sequences, yet Supp Table 1 mentions number of SNPs. It is very important to distinguish SNPs from the number of nucleotides actually examined, because this controls ascertainment effects, and the calibration of N_e . Without it one has no direct estimate of a baseline $\theta = 4N_{ref}\mu$ (Markovtsova et al, 2001, Excoffier et al 2013)

Related to this, the advantage of simulation-based inference (i.e. their use of ABC/CNNs etc) over direct SFS methods will only come from using haplotype information. Otherwise, standard composite likelihood SFS (dadi, fastsimcoal, momi etc), obtaining confidence intervals through bootstrapping, uses pretty much all the information in genomic data pretty well. It looks as though the data here are basically SNPs, maybe some linked together using `de_novo_map.pl`. But the fact that there is not a genome to align against for many of these taxa shows that the authors are really braving the edge of what can be reliably done.

Markovtsova, L., Marjoram, P. & Tavaré, S. (2001). On a test of Depaulis and Veuille. *Molecular Biology and Evolution* 18, 1132–1133.

Excoffier, L., Dupanloup, I., Huerta-Sánchez, E., Sousa, V.C. and Foll, M., 2013. Robust demographic inference from genomic and SNP data. *PLoS genetics*, 9(10), p.e1003905.

Specific points

122-130. Although the authors point to supp table 1 here

157-167 More information in the methods to clarify that monomorphic sites were considered appropriately, during filtering etc, so that the sfs is not biased.

An additional point is that a caveat ought to be given that population structure has a potentially strong confounding effect on these population trajectory estimates (e.g. Loog, 2021; Mazet et al, 2016)

Loog, L., 2021. Sometimes hidden but always there: the assumptions underlying genetic inference of demographic histories. *Philosophical Transactions of the Royal Society B*, 376(1816), p.20190719.

Mazet, O., Rodríguez, W., Grusea, S., Boitard, S. and Chikhi, L., 2016. On the importance of being structured: instantaneous coalescence rates and human evolution—lessons for ancestral population size inference?. *Heredity*, 116(4), pp.362-371.

424-442. The use of simulation-based inference here is primarily for model choice. The authors rightly point to potential issues with choice of summary statistic, which has motivated their approach, although I think there is broad agreement in the statistical community that most of the original bias that was identified arose from using a sparse set of summaries, and subsequent focus has been on efficient use of multiple summaries (Marin et al, 2018).

However, a more pertinent issue here is the sensitivity of Bayesian model choice, generally, to the prior assumed for the underlying parameters within the respective models (Gelman et al, 2013). Have the author explored this? And what is the effect?

Gelman, A., Carlin, J.B., Stern, H.S., Dunson, D.B., Vehtari, A. and Rubin, D.B., 2013. *Bayesian data analysis*. CRC press.

Marin, J.M., Pudlo, P., Estoup, A. and Robert, C., 2018. Likelihood-free model choice. In *Handbook of Approximate Bayesian Computation* (pp. 153-178). Chapman and Hall/CRC.

Reviewer #3:

Remarks to the Author:

This manuscript uses population genetics and palynological records to infer the demography of 5 steppe species from a broad range of sites across Europe. The goal is understanding when species colonized zonal and extrazonal steppe regions and how species dispersed within and between those environments during this process. Overall I think this is an excellent paper. It's well written with clear hypotheses and goals. I think the questions asked are important, the data sets they collect appears large and appropriate for the question (to my understanding, see caveat below), and the methods use both canonical approaches (e.g. STRUCTURE, BPP, Stairway plot) and clever new deep learning + ABC methods that are right cutting edge of the field. Below I list a few comments, all fairly minor, with the hope of shoring up a few places in an already excellent paper.

Comments

- I couldn't find a code repository. This would be particularly useful to have for 1) the code that processes read data to SNPs, and 2) the CNN+ABC method. Though ideally code for other steps would be available as well so the entire data analysis could be reproduced. On the other hand, Supplementary Data 1 gives an excellent summary of where to find the primary sequence data.

-CNN-ABC model calibration. The authors describe posterior probabilities for demographic models on line 177 and in Supp Table 3. To my reading they did cross-validation to assess model accuracy

(i.e. does model select correct demographic scenario?), but how do we know if the posteriors are well calibrated? A model can be accurate but poorly calibrated, and I'd recommend that the authors do a calibration study using the test set data. Here is a decent description of the procedure <https://machinelearningmastery.com/calibrated-classification-model-in-scikit-learn/>. Without assessing calibration, the posteriors are not very meaningful. But if we know this model is well calibrated, then saying all lineages had an explanatory demographic model with a posterior > 0.87 has real meaning.

-Add a diagram of the CNN-ABC method. Canonical representation like: https://www.researchgate.net/figure/Schematic-diagram-of-a-basic-convolutional-neural-network-CNN-architecture-26_fig1_336805909. I'd consider this nice to have, but not essential. If the authors wish to persuade others to use this method, a figure which clearly shows how it is put together may help.

Caveat

-I have expertise in population genetics and machine learning. I was able to carefully evaluate those sections of the MS. However, I have no expertise with the species studied or this steppe ecosystem. Likewise no expertise with interpreting pollen records. So while I read those sections carefully and found nothing that struck me as unusual, I feel I am not in a position to evaluate those sections as rigorously. That said, I feel the authors interpretation of the history of these species is consistent with the population genetic inferences they make.

Point-by-point response to the reviewers

Manuscript: Congruent evolutionary responses of European steppe biota to late Quaternary climate change: insights from convolutional neural network-based demographic modeling

Corresponding authors:

Philipp Kirschner (philipp.kirschner@gmail.com)

Peter Schönswetter (peter.schoenswetter@uibk.ac.at)

Reviewer #1 (Remarks to the Author):

This is an excellent manuscript about the evolutionary responses of the European steppe biota to the Pleistocene climate changes which, as evidenced by the results, have produced deep effects on this biome. Given the relevance of the issue, and the conclusions presented, the ms. merits publication in Nature Communications. In addition, this ms. is complementing well a first paper also published in Nat. Comm. about its conservation value.

In my opinion, the ms. is novel and well-focused, with appropriate methods, and sound, well-discussed results. Albeit I am not familiar with the CNN methodology, the ABC is well implemented, with appropriate pre-defined scenarios. In addition to provide a few minor comments, my only major comment is concerning the distribution models (see below).

AUTHORS: Thank you for the kind words. We respond below one-by-one to the comments.

1. Lines 122-130. These lines are clearly Material & Methods, not Results.

AUTHORS: Agreed. We moved these lines to the M&M section.

2. Lines 227-231. I agree that small population sizes is a factor that can explain the lack of range expansion in the extrazonal lineages. The center-periphery hypothesis (CPH) predicts small population sizes for edge, peripheral populations.

AUTHORS: We fully agree with the reviewer that the CPH should be mentioned in this context.

We correspondingly changed the following passage:

“Smaller population sizes (as predicted by the center-periphery hypothesis for peripheral populations²⁹) and stronger substructuring of source populations previous to expansion, as well as the presence of mountain barriers preventing effective dispersal³⁰⁻³³ were likely key factors that hindered range expansion and subsequent increases in effective population size in the extrazonal lineages. but less so in the zonal lineages.”

3. Lines 351-353. Although the genetic structure will likely not change, I suggest to increase the number of replicates.

AUTHORS: Done. We ran the analyses for 1,000,000 generations, discarding 100,000 generations as burnin and 20 replicate per K. As expected by the reviewer, the revealed genetic structure did not change. This was correspondingly changed in the Materials and Methods. While the STRUCTURE results did not change, all pie charts in Figure 2 were rearranged to render population specific results more visible.

4. I am curious why you used only the LGM time slice to build the distribution models, as in recent times simulated paleoecological data for other Pleistocene time periods are available. These would be useful given the divergence time estimations between lineages. For example, data hosted on PaleoClim (<http://www.paleoclim.org/>) allow you to reconstruct the paleodistribution for the Last Interglacial and MIS19 (ca. 0,79 Mya), while with Oscillayers (<https://doi.org/10.5061/dryad.27f8s90>), time slices can be selected every 10,000 years.

AUTHORS: We indeed considered projecting niche models to multiple time slices at an early stage of the project, but finally decided to incorporate niche models for the LGM only. We refrained from using time slices representing Pleistocene warm stages such as the last interglacial or MIS19 as such models could lead to wrong conclusions for the reasons elaborated below:

Modelling occurrence probabilities of steppe biota using climatic data from the LGM time slice is straightforward because cold stage steppes in Europe were “climatic” or zonal steppes, whose occurrence was driven by the macroclimate (that prevented closed forest establishment) - similar to today’s zonal steppes of inner Eurasia. In other words, the climatic niche under cold stage conditions directly shaped the actual distribution of steppe biota, and biotic interactions were less important.

This stands in marked contrast to models based on warm stage climatic data. The occurrence of steppe biota in today’s warm stage climate is determined by habitat structure – that is open, forest-free steppic grassland – and not by the macroclimate. This means that in today’s temperate climate, extrazonal steppes are often small and always embedded in a matrix of forests. However, the here presented niche models were all exclusively informed by climatic data, and factors such as forest cover or soil depth could neither be incorporated at the given scale, nor is such information available for other time slices. While the predicted niches for present-day conditions definitely reflect continentally influenced areas that have a generally suitable climate for steppe biota, these species’ actual occurrence within this climatic niche ultimately depends on biotic interactions (i. e. the lack of forest and the presence of steppic grassland).

Based on the known actual distributions of steppe biota in Europe, it is evident that the present-day niche models are an overprediction of the actual distribution of steppe biota in Europe (see the present-day models in Kirschner, Zaveska et al. 2020). For the elaborated reasons, this limitation similarly applies when modelling niches of steppe biota for other Pleistocene warm stages that are analogues of today’s temperate forest climate, such as MIS19 or the last interglacial, when forests were the dominant vegetation in Europe as well.

We agree with the reviewer that this needs some clarification, as potential readers might be

similarly curious. We added some lines to the Discussion summarizing the above considerations. In case the reviewer feels that we better show models from multiple time slices, we can do such analyses, but we think that this will not add significant value to the manuscript.

The corresponding section in the Discussion now reads:

“Lineage distribution models for LGM conditions suggested large and continuous suitable habitats for both extrazonal and zonal lineages (Figure 2B). Given that steppes were the zonal – that is, microclimatically driven – vegetation under cold stage conditions, climate-based niche models likely well reflect the species’ actual ranges at the LGM. This is less the case for niche models inferred for present-day, warm stage conditions. While present-day models are certainly restricted to areas with at least moderately continental climate¹⁹, the actual occurrence of steppes within these modelled niches is largely determined by biotic interactions, specifically the lack of a dense forest cover. Modelling biotic interactions has proven problematic at the available spatial resolution⁴⁴; we thus refrained from directly comparing the extent of warm stage and cold stage niches in the context of demography and lineage formation.”

Reviewer #2 (Remarks to the Author):

This paper provides an interesting further set of analyses, based on data generated by Kirschner et al, previously published in Nature Communications. The scope of the project is ambitious: different taxa from a biome, with the aim of understanding the demographic responses to past fluctuations in climate.

The analyses presented in this paper use fairly recently developed approaches in population genomic analysis. I have to say, from a genomic perspective, although the ambition is impressive, the actual data are relatively limited, and I have some doubts about aspects of these analyses.

AUTHORS: Thank you for your comprehensive comments. We are addressing all raised points below.

I note that there is very little direct information in the main text, or the methods, about the extent of the data available. This is given in Supp Table 1. I suggest that it should be more upfront. To augment this, I would think some information on contig/scaffold lengths should be made available. For example BPP, as I understand it, is designed for short non-recombining sequences, yet Supp Table 1 mentions number of SNPs. It is very important to distinguish SNPs from the number of nucleotides actually examined, because this controls ascertainment effects, and the calibration of N_e . Without it one has no direct estimate of a baseline $\theta = 4N_{ref}\mu$ (Markovtsova et al, 2001, Excoffier et al 2013)

Markovtsova, L., Marjoram, P. & Tavaré, S. (2001). On a test of Depaulis and Veuille. *Molecular Biology and Evolution* 18, 1132–1133.

Excoffier, L., Dupanloup, I., Huerta-Sánchez, E., Sousa, V.C. and Foll, M., 2013. Robust demographic inference from genomic and SNP data. *PLoS genetics*, 9(10), p.e1003905.

AUTHORS: Done. We were not aware that we have not provided direct information on the length of the RADseq fragments, as we were only citing the study for which these data were generated - thank you for bringing this up. We added this missing information to the Materials & Methods section that now reads:

“The RADseq data analyzed in this manuscript were generated by Kirschner et al.¹⁹ using the original RADseq protocol⁴⁶ with minor modifications⁴⁷. These data consist of 89 base pair single-end sequences that are available from the NCBI short read archive (Supplementary Data 1).”

We are aware that the n of nucleotides examined is important in estimating N_e . This “length of nucleotides examined” value was correspondingly passed to the Stairway Plot software. We now added this value to Supplementary Table 1.

Further, we concur that a clear definition of the data that was used for each analysis is important. While this was done in the Material & Methods section, it has not been rendered clear enough in Supplementary Table 1. We thank the reviewer for pointing this out. Also Supplementary Table 1 has been moved upfront to the main text now, and is now referred to

as Table 1. We correspondingly adapted the table and its caption as shown below:

“Table 1. Overview of the number of individuals (CNN, Stairway Plot, BPP, and STRUCTURE) and the number of single nucleotide polymorphisms (SNPs; CNN, Stairway Plot, STRUCTURE) used for the respective analysis and for each species (details in Material & Methods). L (Stairway Plot) refers to the total number of nucleic sites (both polymorphic and monomorphic sites) from which SNPs were called. BPP analyses were based on full sequences of random subsets of RADseq fragments.”

Species	CNN		Stairway plot		BPP		STRUCTURE	
	Lineage – Individuals – n of SNPs (percentage of missingness across individuals)		Lineage – Individuals / n of SNPs / L		Lineage – Individuals		Individuals / n of SNPs	
Euphorbia seguieriana	ExZon	80	12,125	ExZon	120 / 5623 / 5×10^5	ExZon	15	138 / 30,804
	Zon	135	(15%)	Zon	84 / 9122 / 8.119×10^5	Zon	15	
Omocestus petraeus	ExZon	10	1763	ExZon	24 / 1964 / 1.748×10^5	ExZon	23	158 / 7016
	Zon	10	(42%)	Zon	12 / 1213 / 1.08×10^5	1Zon	10	
Plagiolepis taurica	ExZon	23	12,542	ExZon	64 / 4825 / 4.294×10^5	ExZon	18	142 / 23,825
	Zon	22	(17%)	Zon	29 / 7016 / 6.244×10^5	Zon	15	
Stenobothrus nigromaculatus	ExZon	12	2922	ExZon	16 / 1068 / 0.95×10^5	ExZon	15	97 / 3088
	Zon	6	(41%)	Zon	9 / 1513 / 1.347×10^5	Zon	8	
Stipa capillata	ExZon	102	3813	ExZon	30 / 4943 / 4.399×10^5	ExZon	15	262 / 9073
	Zon	98	(27%)	Zon	56 / 1828 / 1.627×10^5	Zon	15	

Related to this, the advantage of simulation-based inference (i.e. their use of ABC/CNNs etc) over direct SFS methods will only come from using haplotype information. Otherwise, standard composite likelihood SFS (dadi, fastsimcoal, momi etc), obtaining confidence intervals through bootstrapping, uses pretty much all the information in genomic data pretty well. It looks as though the data here are basically SNPs, maybe some linked together using de novo map.pl. But the fact that there is not a genome to align against for many of these taxa shows that the authors are really braving the edge of what can be reliably done.

AUTHORS: The literature comparing the performance of CNNs, which use alignment images, with other model-based methods is very scarce. For approaches using SNPs without haplotype information, results pointed to a high performance of CNNs for model selection, even surpassing ABC approaches based on common Summary Statistics (Fonseca et al. 2021; Perez et al. 2021), though we were not able to find a direct comparison with a method relying on the SFS. Interestingly, in a comparison using haplotype information, Sanchez et al. (2020) pointed to complementary information from CNNs and the SFS, with higher accuracy when they were combined. Here, as we are analyzing 5 species with different amounts of SNPs and variable levels of missing data, CNNs can provide a more flexible solution, which showed a high accuracy in our cross validation procedure (more than 75% correct model assignment in 4 of the 5 species). Therefore, we believe that our approach was able to capture the complexity contained in the analyzed data and to obtain accurate results for model selection.

Fonseca, E. M., Colli, G. R., Werneck, F. P., & Carstens, B. C. (2021). Phylogeographic model selection using convolutional neural networks. *Molecular Ecology Resources*,

<https://doi.org/10.1111/1755-0998.13427>.

Perez, M. F., Bonatelli, I. A. S., Romeiro-Brito, M., Franco, F. F., Taylor, N. P., Zappi, D. C., & Moraes, E. M. (2021). Coalescent-based species delimitation meets deep learning: Insights from a highly fragmented cactus system. *Molecular Ecology Resources*, <https://doi.org/10.1111/1755-0998.13534>.

Sanchez, T., Cury, J., Charpiat, G., & Jay, F. (2020). Deep learning for population size history inference: Design, comparison and combination with approximate Bayesian computation. *Molecular Ecology Resources*, <https://doi.org/10.1111/1755-0998.13224>.

Specific points

122-130. Although the authors point to supp table 1 here

AUTHORS: This comment is unclear.

157-167 More information in the methods to clarify that monomorphic sites were considered appropriately, during filtering etc, so that the sfs is not biased.

AUTHORS: Monomorphic sites were automatically removed by the populations script implemented in the software Stacks (Catchen et al. 2013) when SNPs are exported to variant call format (vcf). These vcf files that were used to infer the site frequency spectra thus contained polymorphic sites only. We clarify this in the Materials & Methods section that now reads:

“These data consist of 89 base pair single-end sequences that are available from the NCBI short read archive (Supplementary Data 1).”

An additional point is that a caveat ought to be given that population structure has a potentially strong confounding effect on these population trajectory estimates (e.g. Loog, 2021; Mazet et al, 2016)

Loog, L., 2021. Sometimes hidden but always there: the assumptions underlying genetic inference of demographic histories. *Philosophical Transactions of the Royal Society B*, 376(1816), p.20190719.

Mazet, O., Rodríguez, W., Grusea, S., Boitard, S. and Chikhi, L., 2016. On the importance of being structured: instantaneous coalescence rates and human evolution—lessons for ancestral population size inference?. *Heredity*, 116(4), pp.362-371.

AUTHORS: Thank you for raising this point. We are aware that population structure could have an effect on population trajectory estimates and, consequently, cite corresponding studies (references 29 - 32). We relied on the results from Bayesian clustering to define entities on the highest hierarchical level for demographic analyses. We agree however, that it might be worth exploring demography at lower hierarchical levels too - even at the site level. We actually aim to do so in more detail for biota of extrazonal steppes in an upcoming study. However, such an approach is beyond the scope of this study, also because we do not have enough samples per site to pursue such analyses yet.

We have added a short sentence in the text, which now reads:

We emphasize that intrapopulation structure may also affect the ability of demographic methods to detect population expansion³⁴.

424-442. The use of simulation-based inference here is primarily for model choice. The authors rightly point to potential issues with choice of summary statistic, which has motivated their approach, although I think there is broad agreement in the statistical community that most of the original bias that was identified arose from using a sparse set of summaries, and subsequent focus has been on efficient use of multiple summaries (Marin et al, 2018).

However, a more pertinent issue here is the sensitivity of Bayesian model choice, generally, to the prior assumed for the underlying parameters within the respective models (Gelman et al, 2013). Have the author explored this? And what is the effect?

Gelman, A., Carlin, J.B., Stern, H.S., Dunson, D.B., Vehtari, A. and Rubin, D.B., 2013. Bayesian data analysis. CRC press.

Marin, J.M., Pudlo, P., Estoup, A. and Robert, C., 2018. Likelihood-free model choice. In Handbook of Approximate Bayesian Computation (pp. 153-178). Chapman and Hall/CRC.

AUTHORS: We agree that a prior sensitivity exercise would be interesting. Indeed, before selecting the prior values of the final dataset shown in our manuscript, we performed a few preliminary runs under slightly different prior values. From these experiments, we observed that even when we change the priors the recovered models usually remained the same, and more interestingly, the stable model always showed the lowest posterior probability. Additionally, in this new version we calibrated our model selection posteriors following the suggestions by Reviewer#3, and the results remain largely congruent. Therefore, our results now point to an accurate and well calibrated model selection, and they remained stable even when different priors were used. However, we believe that extending this to a systematical assessment of prior sensitivity, by comparing the results under thousands of simulations from different prior values for each of the five studied species, is beyond the scope of our manuscript, and would be extremely time and resource consuming.

Reviewer #3 (Remarks to the Author):

This manuscript uses population genetics and palynological records to infer the demography of 5 steppe species from a broad range of sites across Europe. The goal is understanding when species colonized zonal and extrazonal steppe regions and how species dispersed within and between those environments during this process. Overall I think this is an excellent paper. It's well written with clear hypotheses and goals. I think the questions asked are important, the data sets they collect appears large and appropriate for the question (to my understanding, see caveat below), and the methods use both canonical approaches (e.g. STRUCTURE, BPP, Stairway plot) and clever new deep learning + ABC methods that are right cutting edge of the field. Below I list a few comments, all fairly minor, with the hope of shoring up a few places in an already excellent paper.

AUTHORS: Thank you very much for your helpful comments. We hope to sufficiently answer all raised points below.

Comments

- I couldn't find a code repository. This would be particularly useful to have for 1) the code that processes read data to SNPs, and 2) the CNN+ABC method. Though ideally code for other steps would be available as well so the entire data analysis could be reproduced. On the other hand, Supplementary Data 1 gives an excellent summary of where to find the primary sequence data.

AUTHORS: All code and data used in our CNN+ABC approach were deposited in GitHub (https://github.com/manolofperez/CNN_ABCsteppe).

-CNN-ABC model calibration. The authors describe posterior probabilities for demographic models on line 177 and in Supp Table 3. To my reading they did cross-validation to assess model accuracy (i.e. does model select correct demographic scenario?), but how do we know if the posteriors are well calibrated? A model can be accurate but poorly calibrated, and I'd recommend that the authors do a calibration study using the test set data. Here is a decent description of the procedure <https://machinelearningmastery.com/calibrated-classification-model-in-scikit-learn/>. Without assessing calibration, the posteriors are not very meaningful. But if we know this model is well calibrated, then saying all lineages had an explanatory demographic model with a posterior > 0.87 has real meaning.

AUTHORS: We calibrated the trained model by using the test set and a temperature scaling approach. This method showed a better performance and is more appropriate for multiclass (non-binary) classification tasks than the isotonic regression implemented in the sklearn CalibratedClassifierCV method (Guo et al., 2017).

-Add a diagram of the CNN-ABC method. Canonical representation like: https://www.researchgate.net/figure/Schematic-diagram-of-a-basic-convolutional-neural-network-CNN-architecture-26_fig1_336805909. I'd consider this nice to have, but not essential. If the authors wish to persuade others to use this method, a figure which clearly shows how it is put together may help.

AUTHORS: Thank you for the suggestion. We decided to add a graphic representation

of our CNN-ABC approach as a new figure (Figure 3).

Caveat

-I have expertise in population genetics and machine learning. I was able to carefully evaluate those sections of the MS. However, I have no expertise with the species studied or this steppe ecosystem. Likewise no expertise with interpreting pollen records. So while I read those sections carefully and found nothing that struck me as unusual, I feel I am not in a position to evaluate those sections as rigorously. That said, I feel the authors interpretation of the history of these species is consistent with the population genetic inferences they make.

Reviewers' Comments:

Reviewer #1:

Remarks to the Author:

The authors have correctly addressed all my (minor) concerns; therefore I suggest publication.

Reviewer #2:

Remarks to the Author:

I have read through the revised manuscript, and the authors' responses to reviews. I am satisfied that my comments have been addressed.

Concerning my unfinished specific comment that confused the reviewers, I have no idea. Perhaps it is a palimpsest from something I then revised.

Personally, I would hope that the authors take the opportunity to publish the reviewer comments and their rebuttals, so that readers can get a rounded view.

Reviewer #3:

Remarks to the Author:

The authors have addressed each of my comments. I have no further comments. I think this manuscript will make an excellent addition to the literature.